# Black Tea Reduces Diet-Induced Obesity in Mice via Modulation of Gut Microbiota and Gene Expression in Host Tissues

**DOI:** 10.3390/nu14081635

**Published:** 2022-04-14

**Authors:** Xuanli Liu, Gaosheng Hu, Anhua Wang, Guoqing Long, Yongcheng Yang, Dongdong Wang, Nanfang Zhong, Jingming Jia

**Affiliations:** School of Traditional Chinese Pharmacy, Shenyang Pharmaceutical University, Wenhua Road 103, Shenyang 110016, China; liuxuanlijj@126.com (X.L.); hugsh_2011@163.com (G.H.); sywanganhua@163.com (A.W.); sylongguoqing@163.com (G.L.); 15241821063@163.com (Y.Y.); dongdongwangsy@163.com (D.W.); a1321729686@163.com (N.Z.)

**Keywords:** black tea, obesity, gut microbiota, tissue gene expression, DNA methylation, imprinted genes

## Abstract

Black tea was reported to alter the microbiome populations and metabolites in diet-induced obese mice and displays properties that prevent obesity, but the underlying mechanism of the preventative effect of black tea on high-fat diet (HFD) induced obesity has not been elucidated. Epigenetic studies are a useful tool for determining the relationship between obesity and environment. Here, we show that the water extract of black tea (Lapsang souchong, LS) reverses HFD-induced gut dysbiosis, alters the tissue gene expression, changes the level of a major epigenetic modification (DNA methylation), and prevents obesity in HFD feeding mice. The anti-obesity properties of black tea are due to alkaloids, which are the principal active components. Our data indicate that the anti-obesity benefits of black tea are transmitted via fecal transplantation, and the change of tissue gene expression and the preventative effects on HFD-induced obesity in mice of black tea are dependent on the gut microbiota. We further show that black tea could regulate the DNA methylation of imprinted genes in the spermatozoa of high-fat diet mice. Our results show a mechanistic link between black tea, changes in the gut microbiota, epigenetic processes, and tissue gene expression in the modulation of diet-induced metabolic dysfunction.

## 1. Introduction

Lapsang souchong (LS) is a famous traditional Chinese black tea exclusively produced in the Wuyi (Bohea) Mountain area, Southeast China from the fresh local tea species leaves (*Camellia sinensis* var. *sinensis* cv. Bohea) [1]. Black tea has been explored in recent years for its potential health benefits; obesity prevention, metabolic syndrome alleviation, colitis prevention, and cancer prevention are among them [2,3]. Previous studies have shown that flavonoids isolated from black tea inhibit the development of atherosclerosis in animal models [4]. In addition, theaflavins and thearubigins isolated from black tea alter the microbiome populations and metabolites in mice with diet-induced obesity [5,6]. Several animal studies showed that black tea extract alleviates insulin resistance and regulates the process of bile acid metabolism and reduction of body weight following a high-fat diet [7,8,9]. The mechanisms underlying these effects, however, are unknown and more research is needed to better understand how black tea affects body weight and obesity-related illnesses.

Obesity is a major public health problem across the world and is strongly associated with various chronic diseases, including insulin resistance, type 2 diabetes (T2D), fatty liver disease, and cardiovascular disease [10,11,12]. Variations in obesity prevalence are influenced predominantly by biological, environmental, and behavioral factors, including genetics, medications, gut microbiome, and particularly increased calorie intake and reduced physical activity [13]. In recent years, as interest has increased in epigenetics, epigenetic studies offer a useful tool for understanding the relationship between obesity and environment [14].

In 1942, Conrad Waddington defined epigenetics as heritable changes in gene activity without including deoxyribonucleic acid (DNA) modifications [15]. Most of the known epigenetic processes involve histone modifications, DNA methylation, and non-coding RNA. Diet and environmental variables influence these alterations, which are reversible [14,16,17]. Epigenetic treatment has been found to be effective in treating a variety of physiological and pathological processes in humans, including inflammation, carcinogenesis, and the immunological response [17]. A number of studies showed that the occurrence and inheritance of obesity is related to the influence of diet on epigenetic changes [14]. Epigenetic modifications are sensitive to the availability of endogenous metabolites. For example, increased availability of acetyl-coenzyme A (acetyl-CoA) can increase histone acetyltransferase (HAT) activity [18] and vitamin C metabolism is able to induce changes in DNA methylation [19,20]. Host epigenetic programming in a variety of tissues has been proved to be affected by gut microbial metabolites that are absorbed and metabolized by the host [21]. We hypothesized that the regulatory effect of gut microbiota on epigenetic processes could be significant in the treatment of obesity by prebiotics.

In this study, we find that the water extract of LS reverses HFD-induced gut dysbiosis, alters the tissue gene expression, changes the level of a major epigenetic modification (DNA methylation), and prevents obesity in HFD-fed mice. Our results indicate that the LS-induced change of tissue gene expression and the prevention of HFD-induced obesity in mice were dependent on the gut microbiota. Moreover, LS could regulate the DNA methylation of imprinted genes in the spermatozoa of high-fat diet mice. The results of this study show a mechanistic link between black tea, changes in the gut microbiota, epigenetic processes, and tissue gene expression in the modulation of diet-induced metabolic dysfunction.

## 2. Materials and Methods

### 2.1. Animal Study

Animal experiments were performed according to the approved protocols of the Animal Ethics Committee, Shenyang Pharmaceutical University, China, under permit no. SYPU-IACUC-S2019-05.20-102. C57BL/6J male mice were purchased from Beijing HFK Bioscience Co., Ltd., (Beijing, China). All the mice were kept under controlled light conditions (12 h light/dark cycle) with free access to food and water and maintained in a specific-pathogen-free (SPF) environment. Six-week-old male mice were randomly distributed into ten groups, cohoused by groups that contained six animals each. Mice were fed with either a standard chow diet (1022, HFKBio, 4.0% kcal fat, 18.0% protein) or a high-fat diet (H10141, HFKBio, 41% kcal fat, 43% carbohydrate, 17% protein) for 18 weeks. In the LS tea intervention study, the LS treatment groups were intragastrically administered at doses of 760 mg/Kg per day. All the mice were raised with free access to control chow/HFD and water/antibiotic cocktail. Mean energy intake was recorded once a week for 18 weeks.

In the antibiotic cocktail treatment study, to deplete the gut microbiota, mice were treated with a cocktail of antibiotics for 18 weeks from 6 weeks of age. The antibiotic cocktail consisted of ampicillin (1 g/L), vancomycin (0.5 g/L), neomycin (1 g/L), and metronidazole (0.25 g/L) in autoclaved water. This was changed every 3 days to deplete the gut bacteria [22].

### 2.2. Extraction of LS

Briefly, 300 g LS tea was extracted with 30 L water under reflux for 30 min. The procedure was repeated two more times and all water extracts were combined and evaporated to dryness. Dissolving 115 g of tea dried extracts in 1500 mL pure sterilized water yielded the LS tea infusions.

### 2.3. Preparation of Alkaloids, Polyphenols, and Crude Tea Polysaccharides in LS

300 g LS tea was extracted with 30 L water under reflux for 30 min. The procedure was repeated two more times and all water extracts were combined and concentrated to a final volume of 3 L. The solution of water extracts of LS was extracted with chloroform to enrich alkaloids three times. Water phase was extracted with ethyl acetate to enrich polyphenols three times. Chloroform phase and ethyl acetate phase were combined and evaporated to dryness, respectively. The rest of the water phase was extracted with *n*-butyl alcohol to remove tea pigments. The water solution of the tea extract was precipitated with triple volumes of 95% ethanol and kept at 4 °C overnight to collect the crude tea polysaccharides. After centrifugation at 3000× *g* for 15 min, the precipitates were dissolved in distilled water and the dissociative protein was removed by the Sevag method. The alkaloids, polyphenols, and crude tea polysaccharides infusions were prepared by dissolving 7.5 g, 36.75 g, and 7.8 g of dried extracts with 1500 mL pure sterilized water respectively.

### 2.4. Fecal Microbiota Transplantation

ConvD HFD mice were generated by colonizing HFD feeding C57BL/6J mice with fresh stools obtained from ConvR HFD-LS mice in the fecal microbiota transplantation investigation. Briefly, 6-week-old male donor mice were fed with HFD and intragastrically administered with the LS tea infusions for 4 weeks. Stools were collected and pooled daily under a laminar flow hood in sterile conditions. 100 mg of stools was resuspended in 1 mL of sterile saline and the solution was vigorously mixed for 1 min and centrifugation at 3000× *g* for 5 min. The supernatant was collected and used as transplant material. Fresh stool inocula were prepared on the same day within 10 min before transplantation to prevent changes in bacterial composition. Six-week-old HFD feeding C57BL/6J male mice were inoculated with 0.1 mL of stool inocula per day via oral gavage for 18 weeks, before being killed for subsequent analysis.

### 2.5. Biochemical Analyses

Hepatic lipids were extracted by the Bligh and Dyer extraction method [23]. Briefly, 30 mg frozen liver tissue was homogenized using 1.2 mL PBS. Total lipids were extracted from the liver homogenate in 1.2 mL methanol–chloroform (2:1). The organic extract was dried and reconstituted in 100 μL 10% Triton X-100 in isopropanol. The hepatic levels of total cholesterol (TC) and triglyceride (TG) were measured using Elisa kits (Nanjing Jiancheng Bioengineering Institute, Nanjing, China) according to the manufacturer’s instruction. Total protein was quantified using the Bradford Assay.

### 2.6. Oil Red O Staining

After 20 min of staining with Oil Red O (Sigma, Saint Louis, MO, USA), frozen sections of liver tissue (6 mm thick) were counter-stained with haematoxylin for 1 min after washing with water. Sections were examined under light microscopy.

### 2.7. HE Staining

Sections of epididymal adipose tissues (8 mm thick) were stained with haematoxylin and eosin. Adipocyte minor axis, major axis, and area were analyzed using the Image J software.

### 2.8. Oral Glucose Tolerance Test

Mice were fasted overnight for 12 h and given glucose by oral gavage (2 g/kg BW). Blood glucose was determined with a glucose meter (Roche Diagnostics, Basel, Switzerland) using blood collected from the tip of the tail vein at 0, 15, 30, 60, and 120 min after glucose was intragastrically administered.

### 2.9. 16 S rRNA Gene Sequencing

Genomic DNA was extracted from feces using Stool DNA Kit (Omega, Norcross, GA, USA), following the manufacturer’s instructions. The purity and quality extracted DNA were measured using 0.8% agarose gel electrophoresis.

PCR amplification of the bacterial 16S rRNA gene V3-V4 region was performed using the forward primer (5′-ACTCCTACGGGAGGCAGCAG-3′) and the reverse primer (5′-GGACTACHVGGGTWTCTAAT-3′). Sample-specific 10 bp barcodes were incorporated into the 5′ end of the forward and reverse primers for multiplex sequencing. The PCR was carried out on a Mastercycler Gradient (Eppendorf, Hamburg, Germany) using 25 μL reaction volumes, containing 12.5 μL of Taq PCR MasterMix (2×), 3 μL of BSA (2 ng/μL), 2 μL (5 uM) of each forward and reverse primer, and 7.5 μL of ddH_2_O. Thermal cycling consisted of initial denaturation at 94 °C for 5 min, followed by 28 cycles consisting of denaturation at 94 °C for 30 s, annealing at 55 °C for 30 s, and extension at 72 °C for 65 s, with a final extension of 7 min at 72 °C. PCR amplicons were purified using Agencourt AMPure XP Kit (Beckman, Brea, CA, USA) and quantified using Real Time PCR. Deep sequencing was performed using the IllluminaMiSeq platform at Allwegene Technologles Co., Ltd., (Beijing, China). Image analysis, base calling, and error estimation were performed using Illumina Analysis Pipeline Version 2.6.

Qualified reads were separated using the sample-specific barcode sequences and trimmed with Illumina Analysis Pipeline Version 2.6. The low-quality sequences were filtered using the following criteria: sequences that had a length of <230 bp, sequences that had average Phred scores of <20, and sequences that contained ambiguous bases or did not exactly match to primer sequences and barcode tags. The remaining high-quality sequences were clustered into operational taxonomic units (OTUs) at a similarity level of 97% using Uparse algorithm of Vsearch (v2.7.1) software. OTU taxonomic classification was conducted into different taxonomic groups against the SILVA128 database by the Ribosomal Database Project (RDP) Classifier tool. QIIME (version 1.8.0) was used to generate rarefaction curves and to calculate the richness and diversity indices based on the OTU information.

### 2.10. RNA-Seq Processing and Analysis

RNA was extracted from flash-frozen post-mortem livers using the TRIzol Reagent (Invitrogen, Carlsbad, CA, USA) and genomic DNA was removed using DNase I (TaKara, Kyodo, Japan). Then, RNA quality was determined by 2100 Bioanalyser (Agilent, Palo Alto, CA, USA) and quantified using a Nanodrop 2000 spectrophotometer. A single-end cDNA library was prepared using the TruSeq mRNA Sample Preparation kit (Illumina, San Diego, CA, USA) according to the manufacturer’s specifications. After being quantified by the QuantiFluor dsDNA System, samples were sequenced with the Illumina HiSeq 6000 instrument at Majorbio Bio-Pharm Technology Co., Ltd., (Shanghai, China). Experiments were performed in biological triplicate.

Reads greater than 48 million 150 bp were sequenced per sample. The raw paired end reads were trimmed and quality controlled by SeqPrep and Sickle with default parameters to eliminate any reads with a quality score <20. Then, clean reads were separately aligned to mouse reference genome with orientation mode using HISAT2. The mapped reads of each sample were assembled by StringTie. The expression level of each transcript was calculated according to the transcripts per million reads (TPM) method. Differential expression (DE) was calculated using the DESeq2 with Q value ≤ 0.05 and |log_2_FC| ≥ 1. In addition, the related functional pathways of DE genes were demonstrated by KEGG pathway analysis at BH- (Benjamini and Hochberg) corrected *p*-value ≤ 0.05.

### 2.11. Reduced Representation Bisulfite Sequencing

DNA was extracted from flash-frozen post-mortem livers and spermatozoa of mice. Concentration and quality of the DNA were assessed by Qubit picogreen (Thermo Fisher Scientific, Waltham, MA, USA) and 1% agarose gel electrophoresis. An amount of 5 μg genomic DNA was digested and purified using MspI and Qiagen Mini Purification kit (Qiagen, Hilden, Germany). Methylated paired-end Illumina adapters were ligated to the ends of the DNA fragments using T4 DNA ligase after adenosine added at the 3′ end; fragments sized 100–200 bp were purified by 2% agarose gel electrophoresis. The purified fragments were treated with sodium bisulfite and then amplified by PCR. Samples were sequenced on NovaSeq 6000 (Illumina, San Diego, CA, USA) at Genergy Biotech (Shanghai, China) Co., Ltd.

Differentially methylated loci (DML) and differentially methylated regions (DMRs) were analyzed by using DSS (Dispersion Shrinkage for Sequencing data). For each CpG site, a difference in methylation value >10% and a posteriori probability of Wald test >0.99 or *p*-value < 0.01 was considered to be a DML. A methylation region was defined as a DMR using the following criteria: the proportion of DMLs in this region was more than 50%, the length of this region was at least 50 bp, and the region contained at least 5 CpG sites. It is defined as a differentially methylated gene (DMG) as one element of the gene overlap with a DMR more than 50%.

### 2.12. Statistical Analysis

GraphPad Prism 8.0 was used to create all bar plots in this study. The unpaired two-tailed Student’s *t*-test was used to compare the two groups’ data sets. One-way ANOVA was used to evaluate data sets with more than two groups. *p* value of 0.05 was considered statistically significant. In Figure 1 and Figure 2, error bars represent SEM. QIIME (v1.8.0) was used to analyze sequence data for the 16S rRNA gene sequencing analysis. Beta diversity analysis was performed to investigate the structural variation of microbial communities across samples using Bray–Curtis distance metrics and visualized via principal coordinate analysis (PCoA). Statistically significant separation between groups was performed by pairwise comparisons using the analysis of similarities (ANOSIM) test. Differentially abundant taxa across groups were performed by LEfSe and Metastats analysis.

## 3. Results

### 3.1. LS Prevents HFD-Induced Obesity in Mice

Conventionally raised (ConvR) male C57BL/6J mice, which were allowed to acquire a microbiota from birth to adulthood, were fed with a high-fat diet for 18 weeks from 6 weeks of age. Compared to control chow feeding, HFD feeding led to significant increases in body weight, hepatic total cholesterol, hepatic triglycerides, epididymal fat accumulation, and signs of non-alcoholic fatty liver disease (NAFLD) such as accumulation of lipid droplets in the liver (Figure 1A–F and Appendix A). In HFD-fed mice, LS supplementation (ConvR HFD-LS) reduced the weight gain, hepatic total cholesterol, and triglyceride levels and fat accumulation (Figure 1A–F and Appendix A). Measurement of mean energy intake (Figure 1G) showed that the calories consumed by ConvR Chow-LS and ConvR HFD-LS mice did not differ significantly from ConvR Chow and ConvR HFD mice, respectively. This implies that the preventative effect of LS on HFD-induced obesity was not due to reduced energy intake.

### 3.2. LS-Mediated Prevention of HFD-Induced Obesity in Mice Are Dependent on the Gut Microbiota

To investigate whether gut microbes affect HFD-induced obesity in mice, we treated HFD-fed mice with a cocktail of antibiotics for 18 weeks from 6 weeks of age (pseudo germ-free, PGF). The antibiotic cocktail consisted of ampicillin (1 g/L), vancomycin (0.5 g/L), neomycin (1 g/L), and metronidazole (0.25 g/L) in autoclaved water. In contrast to ConvR Chow mice, the body weight, hepatic total cholesterol, and hepatic triglyceride levels of PGF HFD mice were not significantly increased. Epididymal fat accumulation and signs of NAFLD were also not increased in PGF HFD mice (Figure 1A–F and Appendix A). This suggested that gut microbiota involved in HFD-induced obesity in mice.

To evaluate whether the gut microbiota of LS-treated animals could improve the condition of HFD-fed mice, we transferred the complete (uncultured) microbiota harvested from ConvR HFD-LS group to HFD-fed mice (conventionalized, ConvD) and examined obesity-related characteristics. Similar to their donors, the mice receiving microbiota from the ConvR HFD-LS group (ConvD HFD) showed lower weight gain, reduced hepatic total cholesterol and triglyceride levels, and smaller epididymal adipocytes compared with the ConvR HFD group (Figure 1A–F and Appendix A). The finding revealed that the positive effects of LS were conveyed through stool transplantation, which indicates that gut microbiota is involved in the preventative effects of LS on HFD-induced obesity in mice.

To assess the effects of LS alone, we compared the obesity-related traits of PGF HFD-LS mice with the PGF HFD group. The obesity-related traits were not significantly changed by LS directly without the involvement of the gut microbiota (Figure 1D–F and Appendix A). Fasting glucose levels and results from the oral glucose tolerance test were consistent with the changes observed for the obesity-related traits described above (Figure 1H and Appendix A). These finding suggest that the preventative effects of LS on HFD-induced obesity in mice are dependent on the gut microbiota.

### 3.3. Alkaloids and Polysaccharides in LS Reduce Obesity

To identify the active components of LS responsible for the anti-obesity benefits, we separated the LS extract into three fractions (black tea alkaloids, black tea polyphenols, and crude tea polysaccharides). Notably, the preventative effects of alkaloids on HFD-induced obesity in mice (ConvR HFD-BTA) were strikingly similar to that produced by the whole water extract of LS (Figure 2A,D–H). Moreover, polysaccharides produced modest anti-obesity effects on body weight, epididymal fat, and accumulation of lipid droplets in the liver in HFD-induced obesity in mice (ConvR HFD-CTP), whereas black tea polyphenols had no such impact (ConvR HFD-BTP) (Figure 2A,D–H). Fasting glucose levels and results from the oral glucose tolerance test were consistent with the changes observed for the obesity-related traits described above (Figure 2B,C). These findings imply that the anti-obesity benefits of LS extract are primarily attributable to its alkaloids fraction, with the polysaccharides fraction playing a secondary role.

### 3.4. LS Reverses HFD-Induced Gut Dysbiosis

We examined the effects of LS on gut microbiota composition by analysis of the 16S rRNA gene sequences of microbial samples isolated from the colon of ConvR Chow, ConvR Chow-LS, ConvR HFD, and ConvR HFD-LS mouse groups. Bray–Curtis distance-based Principal Coordinates Analysis (PCoA) revealed a distinct clustering of intestinal microbe communities for each treatment group (Figure 3A). Pairwise comparisons using the analysis of similarities (ANOSIM) test revealed a statistically significant separation between groups (R > 0, *p* < 0.05) (Appendix A). These results showed that both HFD and LS interventions cause significant changes in the microbiota composition. Moreover, PCoA and taxonomic profiling demonstrated that the microbe communities of ConvD HFD mice were more similar to their donors (ConvR HFD-LS) than to ConvR HFD mice (Appendix A). Each group of LS-treated mice was compared with its diet-matched control, i.e., ConvR Chow-LS versus ConvR Chow and ConvR HFD-LS versus ConvR HFD. Similar changes in relative abundance of different taxonomic levels occurred within the family *Clostridiaceae_1* and the *Turicibacter*, *Vagococcus*, and *Eubacterium coprostanoligenes* genera were enriched in the ConvR Chow-LS and ConvR HFD-LS groups, according to LEfSe analyses (Appendix A).

We used Metastats analysis to identify the bacteria altered by HFD feeding and LS treatment. A total of 309 operational taxonomic units (OTUs) were significantly altered by HFD feeding, of which 124 were increased and 185 were decreased compared with chow-fed mice. (Appendix A). In HFD-fed mice, the relative abundance of the *Verrucomicrobiae*, *Gammaproteobacteria,* and *Mollicutes* classes was enriched while the relative abundance of the *Bacteroidia* and *Betaproteobacteria* classes was reduced. Supplementation with LS in HFD-fed mice differentially reversed the relative abundances of 56 OTUs altered by HFD feeding, of which 18 were enriched (*Eubacterium coprostanoligenes*, *Lactobacillus gasseri*, *Bacteroidales* S24-7, etc.) and 38 were reduced (*Roseburia* spp., *Escherichia coli*, *Clostridium* sp. ND2, *Lactobacillus casei*, *Eisenbergiella*, *Ruminococcaceae* UCG-009, *Desulfovibrio*, *Akkermansia*, *Lachnospiraceae* UCG-006, *Blautia*, etc.) by LS compared with ConvR HFD mice (Figure 3B,C). This indicated that LS may reverse the gut dysbiosis induced by HFD feeding. In addition, many bacterial species were altered by LS treatment but not by HFD feeding, which indicates that LS may enrich specific bacterial species (Figure 3B,C).

Next, we assessed the functional pathways related to colon microbe communities that are altered by LS. Using the PICRUSt2 algorithm, we investigated microbiota function based on inferred metagenomes. According to DESeq2, 103 gene pathways differed in abundance between ConvR HFD-LS and ConvR HFD mice out of 252 KEGG pathways evaluated. These included pathways relating to lipid metabolism, amino acid metabolism, endocrine and metabolic disease, carbohydrate metabolism, and endocrine system (Appendix A). In the 40 genera most related to functional pathways, *Anaerotruncus*, *Peptococcus*, *Parvibacter,* and *Akkermansia*, which were altered by LS treatment, were related to energy metabolism. *Lachnoclostridium*, *Anaerotruncus*, *Intestinimonas*, *Clostridium_sensu_stricto_1*, *Lactobacillus,* and *Defluviitaleaceae* UCG-011 were associated with lipid metabolism (primary bile acid biosynthesis, steroid biosynthesis, fatty acid degradation, and biosynthesis of unsaturated fatty acids) and endocrine and metabolic disease (type II diabetes mellitus) (Figure 3D). Additionally, abundances of metabolism of cofactors and vitamins, transcription, replication, and repair pathways were significantly altered in LS-treated mice (Figure 3D). The gut microbiota produces a variety of metabolites that affect host physiology and susceptibility to disease, including small organic acids, bile acids, vitamins, and lipids. Collectively, these findings reveal that LS influences the gut microbiota of HFD-fed mice.

### 3.5. Co-Regulation of Hepatic Genes Associates with Gut Microbiota Altered by HFD-Fed or LS Treatment

We used RNA-seq analyses on livers of different groups of mice (ConvR Chow, ConvR HFD, and ConvR HFD-LS) to see how the HFD and LS affected their physiological function. A total of 244 genes were differentially expressed (DE) in ConvR HFD mice, of which 110 were increased and 134 were decreased compared with ConvR Chow mice, as determined by DESeq2 (Figure 4A and Appendix A). This collection of genes was enriched for pathways involved in cholesterol metabolism, fatty acid degradation, insulin resistance, adaptive immunity, SREBP, and PPAR signaling demonstrated by KEGG pathway analysis (Figure 4B and Appendix A). Of the 244 differentially expressed genes, 184 genes altered in HFD-fed mice differentially reversed by LS treatment (Figure 4A and Appendix A). This indicated that LS may reverse the tissue gene expression in mice altered by HFD feeding and the preventative effects of HFD-induced obesity in mice of LS were dependent on the tissue gene expression influenced by LS.

To assess whether microbiota composition structure changes induced by LS intervention affected tissue gene expression, we compared PGF HFD-LS versus PGF HFD and ConvR HFD-LS versus ConvR HFD mice, respectively. Of the 149 differentially expressed genes, 47 were upregulated and 102 were downregulated in PGF HFD-LS mice compared with the PGF HFD group, which was regulated by only LS without the involvement of the gut microbiota. Interestingly, a total of 1115 genes were differentially expressed in ConvR HFD-LS mice compared with ConvR HFD mice, of which 673 increased and 442 decreased. Although a fraction of DE up (Figure 5A) and DE down (Figure 5B) genes are regulated in both PGF and ConvR mice, the fact that ConvR mice have 9.5-fold more total DE genes in response to LS treatment in than PGF mice suggests that gut microbiota are responsible for a substantial portion of the response to LS treatment in liver. This indicated that the preventative effects of HFD-induced obesity in mice of LS were most dependent on the gut microbiota, consistent with the results previously observed (Figure 1D–F and Appendix A). KEGG pathway analysis of unique and overlapping DE genes in PGF and ConvR mice revealed several oppositely regulated pathways as a function of LS (Figure 5C,D). For instance, while the non-alcoholic fatty liver disease (NAFLD) and MAPK signaling pathways are enriched in DE up genes in ConvR mice, they are enriched in DE down genes in PGF mice. The same pattern is observed for pathways involved in fatty acid elongation. Other important differences include DE gene enrichment for processes involved in arachidonic acid metabolism, biosynthesis of unsaturated fatty acids, and primary bile acid biosynthesis in PGF mice only. Moreover, DE down genes were enriched in insulin resistance, steroid biosynthesis, fatty acid biosynthesis, and glucagon signaling pathways and DE up genes were enriched in cholesterol metabolism, type II diabetes mellitus, and PPAR signaling pathways in ConvR mice only (Figure 5C,D), suggesting that LS treatment regulates the above pathways dependent on the gut microbiota. Importantly, genes involved in fluid shear stress and atherosclerosis and steroid hormone biosynthesis were enriched in DE up genes shared by both PGF and ConvR mice (Figure 5C,D). Pathways involved in regulation of lipolysis in adipocytes are significantly enriched in DE down genes shared by both PGF and ConvR mice, revealing that LS treatment regulates host lipolysis in adipocytes irrespective of gut colonization status (Figure 5C,D).

Of the unique DE genes in ConvR mice, gut microbiota altered expression of genes directly regulates host epigenetic programming. For example, SWI/SNF chromatin remodeling complex protein *Smarcd3* was differentially expressed in ConvR mice only and increased by LS treatment. SWI/SNF complex is a kind of ATP-dependent chromatin remodeling complex, which is involved in the regulation of altering histone-DNA contacts and promoting DNA accessibility [24,25]. The expression level of *Srcap,* which is involved in the regulation of DNA methylation in mammalian cells, decreased in LS-treated mice [26]. *Kdm4a* are involved in the regulation of histone post-translational modifications’ (PTMs) addition or removal [27]. A portion of zinc finger protein modulated by gut microbiota may be involved in recruit activating histone lysine-specific methyltransferases (KMTs) and prevent DNA methylation [28,29]. Moreover, a variety of genes that modulate the availability of small molecule metabolite may also affect host epigenetic programming. Choline, betaine, folate, and methionine biosynthesis and metabolism can impact the availability of the methyl donor in epigenetic programming. There were four genes regulated by LS treatment connected to folate modulation, which can affect the availability of the methyl donor SAM for methyltransferases: *Mthfr*, *Gphn*, *Aldh1l2*, and *Mthfd1l*. These findings imply that gut microbial community composition and metabolite production are crucial elements that connect between LS and host epigenetic programming.

### 3.6. LS and HFD Feeding Regulated the DNA Methylation Related to Obesity

Here, we performed reduced representation bisulfite sequencing (RRBS) on livers of different treatment groups (ConvR Chow, ConvR HFD, and ConvR HFD-LS) to investigate the effect of HFD and LS influence on DNA methylation and the relationship between DNA methylation with the genes expression in liver. We identified 397 differentially methylated regions (DMRs) corresponding to 361 unique genes in ConvR HFD mice versus ConvR Chow controls and 219 DMRs with 228 unique genes in ConvR HFD-LS mice versus the ConvR HFD group. In ConvR HFD mice liver, 17 genes DNA methylation changes including *Ube2l3*, *Etv5*, *Lrp1,* and *Nudt3* were determined related to obesity and 11 genes DNA methylation changes including *Exoc312*, *Mst1r*, *Prkch,* and *Arhgap26* were determined related to Type 2 diabetes (T2D) in the previous study. In the ConvR HFD-LS group, 9 genes DNA methylation changes including *Atg4c*, *Ldlr*, *Scarb1,* and *Ankrd11* were related to obesity, and 6 genes DNA methylation changes including *Zmiz1*, *Pde7b*, *Litaf,* and *Maf* were related to T2D (Appendix A). These results suggest that LS prevents obesity and changes epigenetic biomarkers related to obesity or T2D in the liver tissue of HFD-fed mice. DNA methylation in promoter regions is well known to silence genes and gene body methylation is positively correlated with expression. The relationship between DNA methylation with the genes expression in liver were shown in (Figure 6A–D). Interestingly, differentially methylated gene (DMG) level was altered by HFD feeding, but only 16 genes expression were reversed (Appendix A and Figure 6E,F). This result suggests that there are other epigenetic programming influence genes expression except DNA methylation.

### 3.7. LS and HFD Feeding Altered the DNA Methylation Level of Imprinted Genes in Spermatozoa of Mice

Genomic imprinting was recognized as a parent-of-origin allele differential epigenetic modifications that regulate the monoallelic expression. Imprinted genes play important roles in the growth and development of embryos, certain genetic diseases, and cancer [30]. DNA methylation is critical for the silencing of imprinted gene alleles [31]. Although gametic DNA methylation patterns are erased on both parental genomes after fertilization, the methylation status of alleles at DMRs of imprinting control regions (ICRs) is robustly retained during epigenetic reprogramming [30]. To investigate the effects of HFD and LS on the genomic imprinting, RRBS analysis was performed on spermatozoa in following groups: ConvR Chow, ConvR HFD, and the ConvR HFD-LS group. DNA methylation change of 15 imprinted genes induced by HFD feeding was observed, including *Magel2*, *Ctnna3*, *Gab1,* and *L3mbtl1*, that are reported to be linked to the pathogenesis of obesity, high cognitive processes in adulthood, circadian machinery, and growth and development of embryos (Appendix A) [32,33]. Of the 15 DNA methylation changed imprinted genes regulated by HFD feeding, 6 genes were significantly reversed by LS treatment (Appendix A). This result suggests that HFD feeding may influence the growth and development and the adult energy homeostasis of offspring by changing methylation status of imprinting genes and the supplementation of HFD-fed mice with LS may partly prevent the influence of HFD feeding on offspring by altering the epigenetic reprogram in the spermatozoa of mice.

## 4. Discussion

Although earlier research have demonstrated that black tea extract alleviates insulin resistance [7], relieves high-fat diet-induced NAFLD [9], and alters the microbiome populations and metabolites in diet-induced obese mice [5], the relationship between the gut microbiota altered by black tea and obesity has not been explored. In this work, we find that water extract of LS reverses HFD-induced gut dysbiosis, alters tissue gene expression and the epigenetic processes including DNA methylation and histone modifications, and prevents obesity in HFD feeding mice. Our results indicated that the change of tissue gene expression and the prevention effects on HFD-induced obesity in mice of LS were dependent on the gut microbiota. Meanwhile, our data indicated that black tea potentially compensates for the detrimental effects of a paternal high-fat diet on offspring health by regulating the DNA methylation of imprinted genes in the spermatozoa of mice.

The most recent data from the World Health Organization (WHO) states that worldwide obesity has nearly tripled since 1975. More than 1.9 billion adults are overweight and of these over 650 million are obese [34]. Clearly, the current rising incidence of obesity cannot be explained by genetic differences alone. Variations in obesity prevalence are influenced predominantly by the complicated interplay of susceptibility genes with a variety of environmental and behavioral factors [35]. In our research, obesity in inbred mice C57BL/6J induced by HFD feeding described the influence of diets factors on the occurrence of obesity without genetic confounds.

In recent years, many studies have linked dysbiosis in the gut microbiota composition to the onset of obesity and its accompanying metabolic disorders [36]. In a previous study, obesity was not induced by HF/HS feeding in GF mice, which indicated that HF/HS feeding impacted the host metabolic state in a microbiota-dependent manner [21]. As anticipated, obesity was also not induced by HFD feeding in PGF mice in our research (Figure 1A–F and Appendix A). A number of research showed that prebiotics are being studied for the treatment of obesity and associated metabolic disorders [37,38]. Prebiotics are non-digestible food ingredients that restore or maintain the balance in the gut microbiota composition by promoting the growth of specific beneficial bacteria [39]. Although prior research has revealed that black tea alters the microbiome populations and metabolites in diet-induced obese mice, the mechanisms of black tea’s effect on body weight and obesity-related illnesses remain unknown [5,6]. Our observations noted that LS supplements resulted in significant alterations in the gut microbiota of HFD feeding mice and that the anti-obesity benefits of LS are transferrable through fecal transplantation (Figure 1A–F and Appendix A). In addition, we observed that without the involvement of the gut microbiota, LS could not significantly affect obesity-related characteristics in PGF mice (Figure 1D–F and Appendix A). Our results suggest that LS could be employed as prebiotics to prevent obesity and its related metabolic disorders and the prevention effects of LS on HFD-induced obesity in mice were dependent on the gut microbiota.

There are few reports about the effects of alkaloids in black tea on the gut microbiome in previous investigations. However, it is reported that caffeine in Guarana (*Paullinia cupana* Mart.) changed bacterial genus *Lactobacillus* levels in the gut of Wistar rats [40]. In another study, Fubrick tea supplementation attenuated high-fat diet induced obesity by regulating gut microbiota and increased the concentrations of caffeine, theophylline, and theobromine in serum which were positively correlated with an abundance of *norank f Lachnospiraceae* [41]. It is concluded that alkaloids are potential prebiotics in black tea. In our research, alkaloids show the similar preventative effects to that produced by the whole water extract of LS on HFD-induced obesity in mice (Figure 2A–H). The abundance of *Lactobacillus* spp. and *Lachnospiraceae* spp. levels were changed by LS supplement (Figure 3B,C). Dietary polysaccharides are also widely studied as prebiotics for the treatment of diet-induced obesity [42]. Fuzhuan brick tea polysaccharides have been shown to reduce metabolic syndrome in high-fat diet induced mice and modulate the gut microbiota. In our research, crude tea polysaccharides in LS showed modest anti-obesity effects in HFD-induced obesity in mice (Figure 2A–H). Although earlier research have claimed that black tea polyphenols reduce weight gain and alter microbiome populations and function in diet-induced obese mice [5], in our results no significant effect was observed for black tea polyphenols (Figure 2A–H). In another research, black tea polyphenols did not significantly decrease blood glucose levels in db/db mice [43]. Our results suggest that alkaloids are the main key components in LS that have anti-obesity properties. In contrast, crude tea polysaccharides had a minor anti-obesity impact, but black tea polyphenols had no effect.

During the millions of years symbiosing with humans, gut microbiota has developed very complex molecular mechanisms that regulate the host homeostasis. One such mechanism is to influence the host physiological function by regulating the epigenetic processes [44]. Epigenetic alterations are sensitive to the availability of endogenous metabolites and influenced by diet and environmental factors, according to previous research [14,18]. Given the susceptibility of these epigenetic processes to endogenous metabolites, host epigenetic programming in a variety of tissues is proved to be affected by gut microbial metabolites absorbed and metabolized by the host [21]. The short-chain fatty acids (SCFAs) are the main small organic acids produced in the gut by microbes using undigested carbohydrates as substrates [45]. These SCFAs can regulate epigenetic processes by converting to acetyl-CoA, the substrate for histone acetyltransferase (HAT). Moreover, butyrate is a known histone deacetylase (HDAC) inhibitor which can induce Cited2 expression [46]. *Faecalibacterium prausnitizii* and *Roseburia* spp. can produce more than 85% butyrate in the colon [47]. In our research, the relative abundance of *Roseburia* spp. altered by HFD feeding can be reversed by LS treatment (Figure 3B,C). Butyrate can regulate the expression of *Cited2,* which associates with obesity by inhibiting the HDAC [48,49]. In our research, *Cited2* was differentially expressed by LS treatment with the involvement of microbiota. Similarly, mammalian gut microbiota has been shown to manufacture folate [50]. Folate and methionine biosynthesis and metabolism can affect the availability of S-adenosyl methionine (SAM), a methyl donor in epigenetic programming [51]. The intake of folate was prospective adversely linked with the incidence of diabetes in previous studies [52]. It has been shown that levels of folate in the circulation of subjects with T2D have reduced and that this has a positive correlation average degree of DNA methylation in liver biopsies from T2D subjects. In addition, *Bacteroides* spp. significantly improved the signs of NAFLD in the HFD-fed mice by producing folate [53,54]. In our research, the relative abundance of *Bacteroides* spp. enriched by LS treatment in ConvR HFD mice (Figure 3B,C) and four genes (*Mthfr*, *Gphn*, *Aldh1l2*, and *Mthfd1l*) linked to regulation of folate were differentially expressed by LS treatment. Moreover, metabolites of black tea such as the epigallocatechin-3-gallate (EGCG) have also shown the inhibiting DNA methyltransferase (DNMT) activity [54]. It was reported that EGCG inhibits DNA methylation by decreasing SAM [55,56]. We observed that most differentially expressed genes altered by LS treatment in mice liver were dependent on the gut microbiota. Several DE genes (*Smarcd3*, *Srcap,* and *kdm4a*) altered by LS treatment are dependent on the gut microbiota linked to the epigenetic programming include chromatin remodeling, DNA methylation, and histone PTM. Together, these data suggested that gut microbial composition and metabolite production are crucial links between dietary factors and host physiological functions.

DNA methylation is a major epigenetic factor influencing host physiological functions. Several studies have linked obesity to DNA methylation patterns and indicated that differential methylation could predict obesity. In an epigenetic association study, researchers found that *Etv5*, *Nudt3*, *Wwox*, *Zeb1,* and *Maf* DNA methylation alterations were related to obesity in peripheral blood leukocytes of humans. Several genes (*Atg4c*, *Ldlr*, *Scarb1*, *Ube2l3*, *Lrp1*, *Cmip*, *Galnt2*, *St3gal4,* and *Apob*) were identified to be associated with dyslipidemia and *Notch2* DNA methylation changes were identified that associated with diabetes [57]. In another study, it is presented that *Srebf1* is intimately involved in cholesterol biosynthesis and *Srebf1* DNA methylation in leukocytes is associated with BMI [58]. Another study of DNA methylation in whole blood and the association with obesity observed the association between DNA methylation changes of *Ddah2*, *Cux1*, *Notch4,* and *Dst* with BMI [59]. In diabetic islets, adipose, or liver tissue, genes (*Exoc3l2*, *Mst1r*, *Prkch*, *Arhgap26*, *Itga3*, *Pde7b*, *Litaf*, *Galnt10*, *Zmiz1*) DNA methylation changes were identified to be associated with T2D [53,60,61]. Wahl et al. reported that *Ahdc1*, *Gse1,* and *Dlec1* DNA methylation in blood, adipose, and liver tissue associated with BMI and BMI–DNA methylation associations of *Ankrd11* and *Cpsf4l* were identified in blood and adipose tissue. Moreover, BMI–DNA methylation associations of *Snd1* were found in blood. Interestingly, they proved that changes in DNA methylation in tissue are most often the result of obesity, but sometimes also the cause, by using Mendelian randomization [62]. Mendelson et al., also provides evidence for the similar result [63]. We observed that LS treatment altered 13 genes DNA methylation related to obesity or T2D and reversed 16 genes expression of 35 genes DNA methylation level altered by HFD feeding (Appendix A and Figure 6E,F). Our results suggest that LS prevents obesity and changes epigenetic biomarkers related to obesity or T2D in the liver tissue of HFD feeding mice.

In addition to the modulation of the epigenetic modifications in liver tissue, our in vivo experiments demonstrated that the supplement of LS also regulates the DNA methylation status of imprinted genes in mice spermatozoa, e.g., *Gnas*, an important imprinted gene which can affect behavioral timing and response to fear in mice and plays an important role in REM (rapid eye movement) sleep in mice [64]. Moreover, upon losing the paternal *Grb10* expression in the brain, mice are more likely to succeed in a social dominance test and have an increased willingness to wait for a larger reward in a delayed reinforcement task [65,66]. In addition to regulating the high cognitive processes and social behavior in adult animals, LS also modulates the placental and fetal growth patterns and subsequent adult metabolic phenotypes by genomic imprinting. Biallelic expression of *Igf2r* results in a proportional growth reduction in embryos and placenta and the loss-of-function of *Igf2r* is related to ‘large offspring syndrome’ [67]. Mice with a paternal deletion of *Nnat* displayed decreased beta cell storage and secretion of mature insulin, hyperphagia and reduced energy expenditure, resulting in obesity caused by either aging or high-fat diet feeding [68,69]. These results suggest that LS may partly prevent the influence of HFD feeding on offspring by altering the epigenetic reprogram in the spermatozoa of mice.

## 5. Conclusions

In conclusion, our results suggest that LS reduces obesity and diabetes-related symptoms dependent on modulating the composition of the gut microbiota. Our findings also show that LS alters the tissue gene expression and the DNA methylation status in the liver tissue and spermatozoa of mice. Collectively, our results suggest that LS may be used as prebiotics to prevent and treat obesity and the associated metabolic disorders and potentially compensates for the detrimental effects of a paternal high-fat diet on offspring health.

## Figures and Tables

**Figure 1 nutrients-14-01635-f001:**
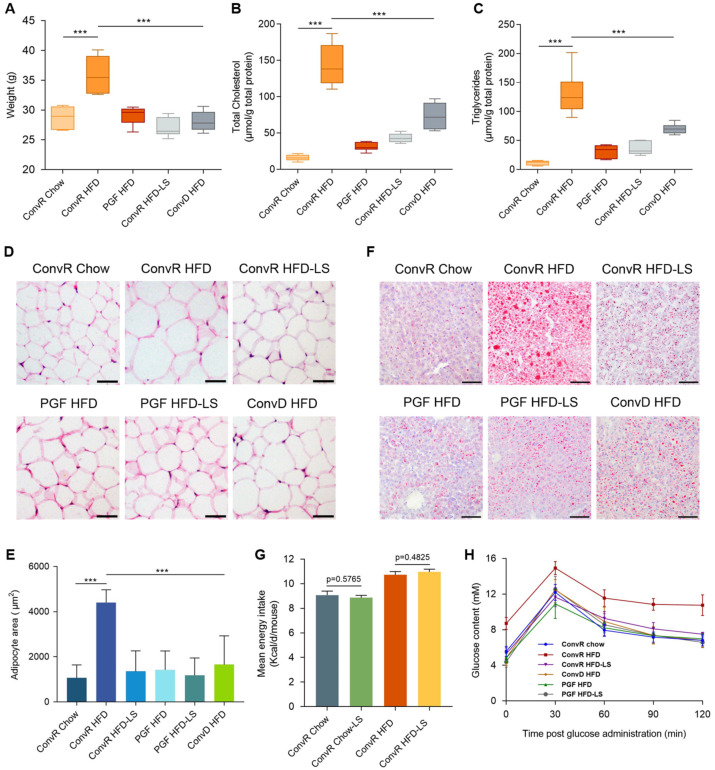
Body weight and lipid lowering effects of LS. Weights (**A**), hepatic total cholesterol (**B**), hepatic triglycerides (**C**), epididymal adipocyte size (**D**,**E**), and liver lipid content (**F**) of Chow- and HFD-fed ConvR, PGF, or ConvD mice treated with water or LS at the time of sacrifice. In (**D**), adipocyte size was estimated using the Image J software. Scale bar, 50 μm. In (**F**), liver lipid content was assessed using oil red O staining. Scale bar, 100 μm. (**G**) the mean energy intake of normal diet or HFD fed mice after tea intervention. Energy intake was determined based on calorie intake from consumed food. (**H**), the oral glucose tolerance of mice with different trentments. Mice fasted overnight were given glucose by oral gavage (2 mg/kg BW). Blood glucose was determined at 0, 15, 30, 60, and 120 min post glucose administration. Data were expressed as mean ± SEM. Differences of data in mice (**A**–**C**,**E**,**F**) were analyzed using one-way ANOVA analysis *** *p* < 0.001; error bars represent SEM; *n* = 6 mice per condition. The mean energy intake (**G**) was analyzed using unpaired two-tailed Student’s *t*-test.

**Figure 2 nutrients-14-01635-f002:**
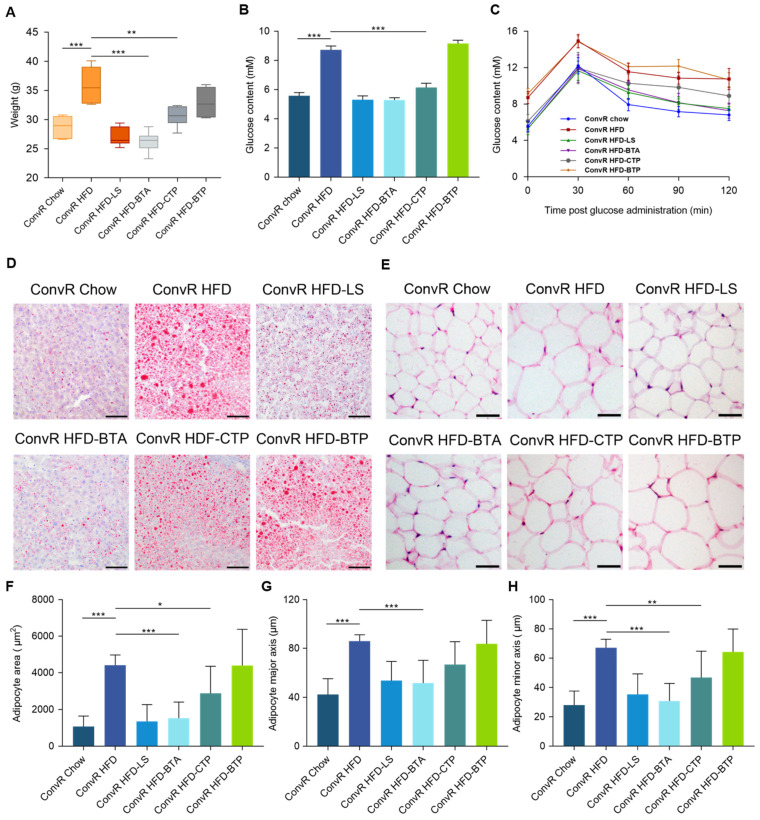
Body weight and lipid lowering effects of BTA, CTP, and BTP. Weights (**A**), liver lipid content (**D**), and epididymal adipocyte size (**E**–**H**) of Chow- and HFD-fed ConvR mice treated with water, LS, BTA, CTP, or BTP at the time of sacrifice. In (**D**), liver lipid content was assessed using oil red O staining. Scale bar, 100 μm. In (**E**), adipocyte size was estimated using the Image J software. Scale bar, 50 μm. Fasting glucose (**B**) and the oral glucose tolerance (**C**) of mice with different trentments. Mice fasted overnight were given glucose by oral gavage (2 mg/kg BW). Blood glucose was determined at 0, 15, 30, 60, and 120 min post glucose administration. Differences of data in mice (**A**–**C**,**F**–**H**) were analyzed using one-way ANOVA analysis *** *p* < 0.001, ** *p* < 0.01, * *p* < 0.05; error bars represent SEM; *n* = 6 mice per condition.

**Figure 3 nutrients-14-01635-f003:**
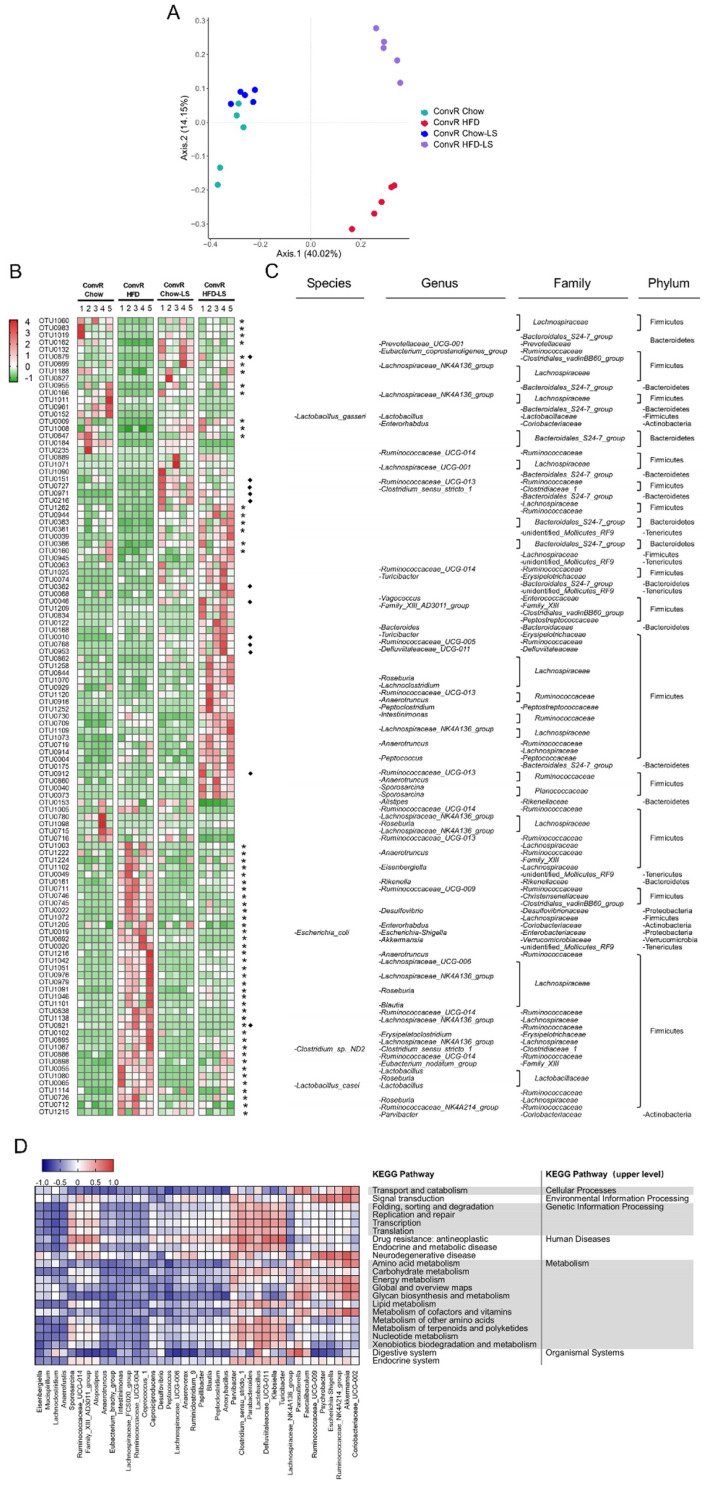
LS alters microbiota composition in HFD-fed mice. (**A**) Principal coordinate analysis (PCoA) plot based on the Bray–Curtis distance OTU matrix of mouse colon microbiota in ConvR Chow, ConvR Chow-LS, ConvR HFD, and ConvR HFD-LS groups. (**B**) Heatmap showing the abundance of 111 OTUs significantly altered by LS in HFD-fed mice based on Metastats analysis. Numerical mapping mode based on Z-score. *n* = 5 mice per condition. Stars represent OTUs whose abundance in chow-fed mice was altered by HFD and then reversed by LS. Diamonds indicate the OTUs that specific bacterial species were enriched by LS. (**C**) Represents bacterial taxa information (species, genus, family, and phylum) of 111 OTUs from B are shown. (**D**) Microbiota composition altered by LS in HFD-fed mice are related to several gene functional pathways. Spearman’s correlation coefficients were estimated for each pairwise comparison of genus counts and KEGG pathway counts. Only KEGG pathways and genera of interest are included in the heatmap.

**Figure 4 nutrients-14-01635-f004:**
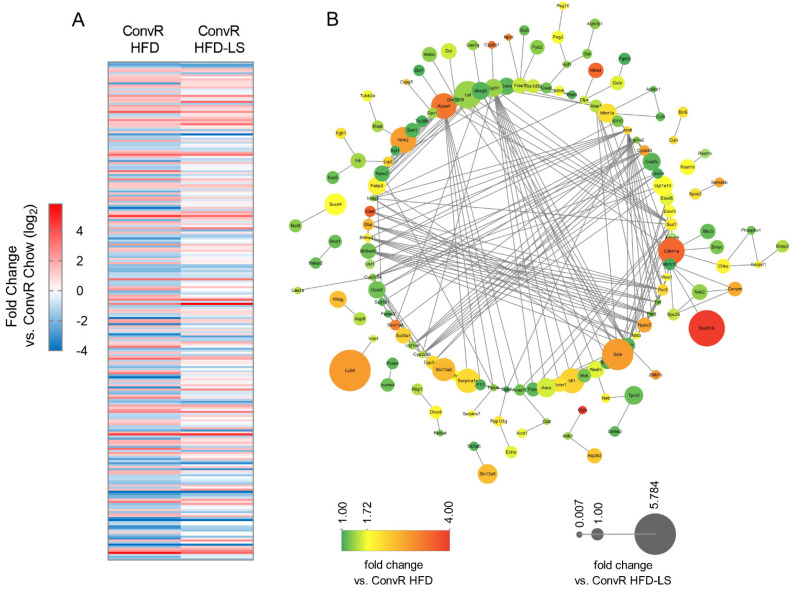
Hepatic genes in ConvR Mice are regulated by HFD fed or LS treatment. (**A**) 244 differentially expressed hepatic genes regulated by HFD fed (ConvR HFD and ConvR HFD_LS fold change versus ConvR Chow, log2). (**B**) Interaction network for 244 differentially expressed hepatic genes regulated by HFD fed. Node color indicates relative expression in ConvR HFD mouse livers. Node size indicates relative expression in ConvR HFD_LS mouse livers. *n* = 3 mice per condition.

**Figure 5 nutrients-14-01635-f005:**
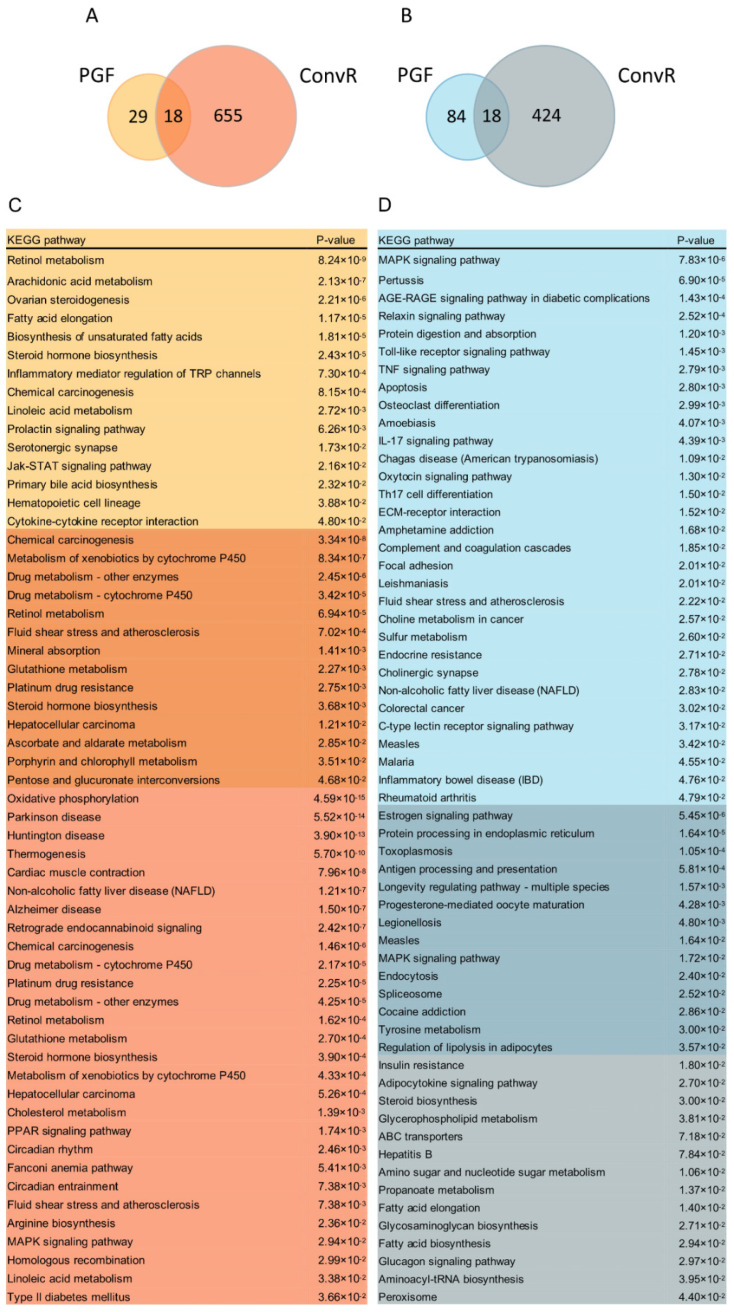
Regulation of hepatic transcription as a function of LS in PGF and ConvR mice. Overlap of hepatic DE genes that significantly (**A**) increase and (**B**) decrease in LS treatment mice vs. HFD-fed controls. Numbers in each portion of the Venn diagram represent the number of corresponding DE genes in each category. (**C**,**D**) KEGG pathway enrichment in DE up genes (**C**), and DE down genes (**D**). Orange represent DE up in PGF mice only, orange/red indicate DE up in both PGF and ConvR mice, red represent DE up in ConvR mice only. Blue indicate DE down in PGF mice only, blue/grey represent DE down in both PGF and ConvR mice, grey indicate DE down in ConvR mice only. *n* = 3 mice per group, significance was determined by DESeq2 with *p*-adjust < 0.05 and |log_2_FC| ≥ 1.

**Figure 6 nutrients-14-01635-f006:**
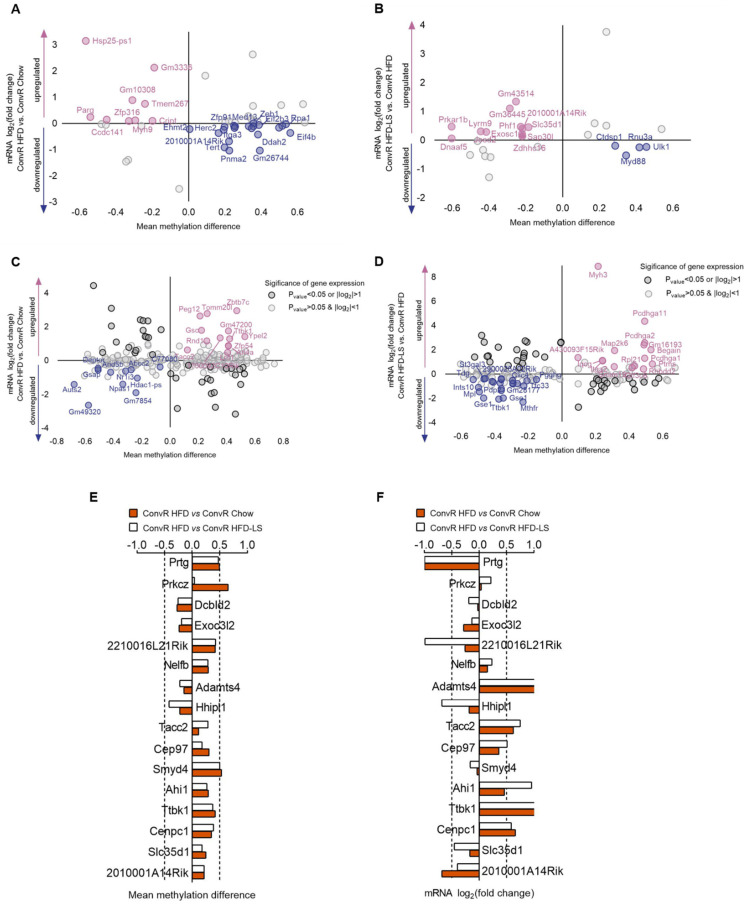
The association of Hepatic gene expression with DNA methylation regulated by HFD fed or LS treatment. (**A**,**B**) the negative correlation between methylation of promoter regions with gene expression in DMGs regulated by HFD fed (**A**) and LS treatment (**B**). (**C**,**D**) the positive correlation between methylation of gene body with gene expression in DMGs regulated by HFD fed (**C**) and LS treatment (**D**). The DNA methylation level (**E**) and genes expression (**F**) of 16 DMGs regulated by HFD fed reversed by LS treatment. *n* = 3 mice per condition.

## Data Availability

The source data underlying Figure 1, Figure 2, Figure 3, Figure 4, Figure 5 and Figure 6 and Appendix A are provided as a Source Data file. The 16S rRNA gene sequences, RNA-seq sequences, and RRBS data were provided and available at NCBI Sequence Read Archive (SRP) repository with BioProject ID PRJNA745524. Other data supporting the findings of this study are available from the corresponding authors upon reasonable request.

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
