# Peer review of "Black Tea Reduces Diet-Induced Obesity in Mice via Modulation of Gut Microbiota and Gene Expression in Host Tissues"

_nutrients, 2022, doi:10.3390/nu14081635_

Round 1
Reviewer 1 Report
This study is very interesting, but I have several concerns.
(1) The authors conclude as follows: “we observed that the obesity-related traits could not be significantly changed by LS directly without the involvement of the gut microbiota in PGF mice (Figure. 1D-F and Supplementary Figure. 2A-C). Our results suggest that LS may be used as prebiotics to prevent obesity and its related metabolic disorders, and the prevention effects of LS on HFD-induced obesity in mice were dependent on the gut microbiota. “ However, since PGF mice do not induce obesity, experiments administering LS to PGF mice are not appropriate.
(2) Although the authors have demonstrated the involvement of gut bacteria in anti-obesity by FMT, to rule out the contribution of gut contents other than gut bacteria, the authors should demonstrate that filtered gut contents that gut bacteria removed show no anti-obesity effect.
(3) To identify the active constituents of LS involved in the anti-obesity effect, the authors fractionated LS extracts into three fractions and examined their contribution to the anti-obesity effect. I think this experiment is excellent, but the authors should show the effect of the three fractions administration on gut bacteria.
Author Response
(1) The authors conclude as follows: “we observed that the obesity-related traits could not be significantly changed by LS directly without the involvement of the gut microbiota in PGF mice (Figure. 1D-F and Supplementary Figure. 2A-C). Our results suggest that LS may be used as prebiotics to prevent obesity and its related metabolic disorders, and the prevention effects of LS on HFD-induced obesity in mice were dependent on the gut microbiota. “ However, since PGF mice do not induce obesity, experiments administering LS to PGF mice are not appropriate.
Response: Thank you for your comments. In our research, Supplementation with LS reduced the weight gain, hepatic total cholesterol and triglyceride levels and fat accumulation in HFD-fed mice, It shows that black tea has a good anti-obesity effect. As has been reported in many previous literatures, the mechanism of anti-obesity is very complex, and drugs can exert anti-obesity effect through many ways, such as directly interacting with the host or indirectly acting through regulating gut microbiota (Jakab et al. Diabet Metab Synd Ob. 2021,14: 67–83 and Kobyliak et al. Nutr Metab. 2016, 13: 14). Therefore, in our study, the anti-obesity effect of black tea also exists in two situations, directly effect on the host, or indirectly effect on the host by regulating the gut microbiota. Pseudo-germfree (PGF) mouse models and fecal microbiota transplantation (FMT) experiments are commonly used to investigate the effects of drugs independently or in dependence on gut microbiota (Qiao et al. Cell Rep. 2020, 32: 108005). To determine whether black tea exerts anti-obesity effects independently of gut microbiota, the black tea treatment PGF HFD mice were compared with PGF HFD mice in our research. The results showed that the body weight, hepatic total cholesterol and hepatic triglyceride levels of PGF HFD-LS mice were not significantly different with PGF HFD mice. These results suggest that the anti-obesity effect of black tea may be indirectly induced by regulating the gut microbiota of mice.
(2) Although the authors have demonstrated the involvement of gut bacteria in anti-obesity by FMT, to rule out the contribution of gut contents other than gut bacteria, the authors should demonstrate that filtered gut contents that gut bacteria removed show no anti-obesity effect.
Response: Thank you for your question. It is a good suggestion to investigate the gut microbe-dependent manner of black tea on obesity by comparing fecal microbiota transplantation with gut bacteria removed feces. We noted that although some literatures treated mice with intestinal contents after removal of gut microbiota as the control group to exclude the effects of intestinal contents, most literatures did not consider the effects of intestinal contents in the experiments of fecal microbiota transplantation (Chang et al. Nat Commun. 2015, 6:7489 and Huang et al. Nat Commun. 2019, 10:4971). To this end, we attempted to analyze the reason. As ConvR HFD-LS mice were used to be the donor, the intestinal contents mice were mainly composed of food and related products, own metabolites of mice, gut microbiota metabolites, black tea and related products. As for the components related to black tea, we believed that after the metabolism and absorption of donor mice, the content of these components in the intestinal contents of donor mice was very low, and only 100mg stool was collected and dissolved in 1ml sterile saline, and finally 0.1ml/ mouse/day is given to recipient mice. It was clear that the amount of these components ended up in the recipient mice was very small (compared to the ConvR HFD-LS mice) and its effect on the results was very limited. Therefore, we think our method is acceptable.
(3) To identify the active constituents of LS involved in the anti-obesity effect, the authors fractionated LS extracts into three fractions and examined their contribution to the anti-obesity effect. I think this experiment is excellent, but the authors should show the effect of the three fractions administration on gut bacteria.
Response: This is a good suggestion, and we have listed this experiment as the main content of our follow-up research. We are also eager to know which fraction or which kind of compounds is mainly responsible for the observed effects of LS extract on gut microbes in HFD mice. This will be discussed in detail in our future studies.
Reviewer 2 Report
Dear Authors,
Regarding the article entitled "Black tea reduces diet induced obesity in mice via modulation of gut microbiota and gene expression in host tissues", I consider that it is correctly addressed, and the experiments are appropriate. However, I have a few comments:
- Why were male mice used? And female?
- In statistical analysis, it should include how the data are given (e.g., mean±SEM or mean±SD)
- In line 309, error bars represent SD or SEM?
- Add deoxyribonucleic acid before DNA the first time it appears in the text.
Author Response
(1) Why were male mice used? And female?
Response: Thank you for your comments. In our study, we investigated the therapeutic effects of black tea on diet-induced obesity and its epigenetic effects. Due to the experimental period, the hormone changes caused by aging in female mice will have an impact on fat metabolism, which is one of the reasons we chose male mice. In addition, in the genomic imprinting study, the sperm cells of mice showed genome-wide high levels of methylation, while the levels of methylation in egg cells were lower, which is another reason we selected male mice for the study.
(2) In statistical analysis, it should include how the data are given (e.g., mean±SEM or mean±SD)
Response: Thank you for your reminding. We have added it in statistical analysis.
(3) In line 309, error bars represent SD or SEM?
Response: We have corrected that mistake. Thank you very much.
(4) Add deoxyribonucleic acid before DNA the first time it appears in the text.
Response: We have revised that question. Thank you for your suggestion.